# The Role of Ryanodine Receptors in Regulating Neuronal Activity and Its Connection to the Development of Alzheimer’s Disease

**DOI:** 10.3390/cells12091236

**Published:** 2023-04-25

**Authors:** Giuseppe Chiantia, Enis Hidisoglu, Andrea Marcantoni

**Affiliations:** 1Department of Neuroscience, University of Turin, 10125 Turin, Italy; 2Department of Drug and Science Technology, University of Torino, Corso Raffaello 30, 10125 Torino, Italy; 3N.I.S. Center, University of Torino, 10125 Turin, Italy

**Keywords:** ryanodine receptors, neuronal excitability, synaptic function, Alzheimer’s disease, calcium dysregulation

## Abstract

Research into the early impacts of Alzheimer’s disease (AD) on synapse function is one of the most promising approaches to finding a treatment. In this context, we have recently demonstrated that the Abeta42 peptide, which builds up in the brain during the processing of the amyloid precursor protein (APP), targets the ryanodine receptors (RyRs) of mouse hippocampal neurons and potentiates calcium (Ca^2+^) release from the endoplasmic reticulum (ER). The uncontrolled increase in intracellular calcium concentration ([Ca^2+^]_i_), leading to the development of Ca^2+^ dysregulation events and related excitable and synaptic dysfunctions, is a consolidated hallmark of AD onset and possibly other neurodegenerative diseases. Since RyRs contribute to increasing [Ca^2+^]_i_ and are thought to be a promising target for AD treatment, the goal of this review is to summarize the current level of knowledge regarding the involvement of RyRs in governing neuronal function both in physiological conditions and during the onset of AD.

## 1. Introduction

Alzheimer’s disease (AD) is characterized by specific hallmarks, including the accumulation of Amyloid Beta (Abeta) oligomers, which are derived from Amyloid Precursor Protein (APP) processing. Their consequent effect on neuronal calcium dysregulation and related synaptic dysfunction has been the subject of numerous investigations in recent years. The uncontrolled increase in [Ca^2+^]_i_ either results from alterations in several mechanisms of Ca^2+^ entry through the plasma membrane or Ca^2+^ release from the endoplasmic reticulum. There is considerable evidence to indicate that Abeta oligomers affect Ca^2+^ entry through N-methyl-D-aspartate receptors (NMDARs) [1,2,3] and/or voltage gated calcium channels (VGCCs) [4,5,6,7]. The alternative mechanism of [Ca^2+^]_i_ increase involves the upregulation of ryanodine (RyRs) and inositol triphosphate receptors (IP_3_Rs) and increasing amounts of evidence suggest that they are impaired during AD onset. These two receptors are not equally distributed along the ER membrane and, while IP_3_Rs are more concentrated at the somatic region, RyRs prefer the portion of ER membrane that is in the vicinity of the synaptic sites [8], suggesting that they are significantly involved in the control of [Ca^2+^]_i_ and directly responsible for governing synaptic function. Three different RyR isoforms are known and, while it was previously thought that RyR_1_ was principally expressed in skeletal muscle, RyR_2_ in the heart, and RyR_3_ in neurons [9], it is now widely accepted that all three isoforms are expressed in the brain [10]. Particular importance has been attributed to RyR_2_ [11] and RyR_3_ [12] in the hippocampus. RyRs are composed of four identical subunits of approximately 5000 amino acids each forming a square around a central pore and often complexed with other accessory proteins, such as calmodulin, calstabin, phosphodiesterases phosphatases, and kinases that together stabilize RyRs and modulate calcium release [13]. RyRs are the largest known Ca^2+^ channels and are characterized by high permeability [14]. It has been estimated that the average conductance of RyR_1_ for calcium is 110 pS and that this can increase to over 700 pS for potassium [14], which is also known to permeate through the pore [15]. Seeing as the ion conductance of most ion channels in the plasma membrane ranges from some single pS to tens of pS [16], the significant contribution of RyRs in governing [Ca^2+^]_i_ becomes evident. The Ca^2+^ released from RyRs can be triggered by Ca^2+^ entry through the Ca^2+^ channels located in the plasma membrane; a phenomenon called “Ca^2+^ induced Ca^2+^ release” (CICR) that is typically observed in cardiac myocytes where RyR_2_ is the principally expressed isoform. VGCCs [15] and NMDARs [3] also activate RyRs in neurons through CICR. Alternatively, RyRs can be mechanically activated by VGCCs through direct interaction [17]. This latter mechanism is typically observed in skeletal muscle where Ca^2+^ is released from RyRs upon membrane depolarization [14]. A growing sum of evidence suggests that these receptors play a fundamental role in governing neuronal function. The purpose of this review is to examine and clarify the role of RyRs in regulating neuronal excitability and synaptic properties. Finally, their possible contribution in triggering the onset of neurodegenerative diseases, such as AD, will be discussed and RyR-targeting-based therapeutic strategies will be described.

## 2. Involvement of RyRs in Synaptic-Function Control

RyRs are clearly involved in long-term synaptic plasticity [18,19,20], but are also engaged in short-term synaptic plasticity with varying degrees of impact, depending on the type of neuron and synapse. Calcium released from RyRs activates the calcium dependent inactivation (CDI) mechanisms of VGCCs in neurons [15,21,22,23]. Although interesting results on thalamocortical neurons [22], have been published and RyRs, but not IP3Rs, have been observed to provide a clear contribution to CDI, no clear data on the contribution of the CDI mechanism, as modulated by calcium released from RyRs, to synaptic function are available. Besides controlling the inactivation mechanisms of presynaptic VGCCs, the calcium released from RyRs can directly contribute to modulating synaptic function. In this regard, significant differences between neurons and synapses can be observed [24,25], together with different mechanisms of neurotransmitter release, represented by synchronous, asynchronous, and spontaneous release [26]. Synaptic release occurs via various vesicle pools, which are all distinguished by different calcium sensors associated with synaptic vesicles [26] and different distances from the calcium source [27]. Calcium sensitivity defines the calcium source responsible for triggering vesicle exocytosis and contributes to further characterizing the physiological properties of the different synapses of central neurons. Spontaneous release is known to be important for neuronal development [28], and for glutamatergic [29] and GABAergic [30] synapse maturation. These quantal events are less reliant, compared to synchronous release, on the input of calcium through voltage-gated calcium channels located at the presynaptic plasma membrane [31], and they can be positively modulated by calcium released from RyRs [32]. Contrasting evidence indicates that, while RyR inhibition in the hippocampus reduces the frequency of mEPSCs (miniature Excitatory Post Synaptic Currents) in CA3 pyramidal neurons both in basal conditions [33] and upon nicotine or caffeine (a RyRs agonist) administration [34], caffeine does not affect the spontaneous release of glutamate in CA1 pyramidal neurons [35]. A different scenario has been depicted when the contribution of calcium released from RyRs in the generation of mIPSCs (miniature Inhibitory Post Synaptic Currents) has been studied, with observation that the administration of nanomolar concentrations of ryanodine in CA3 pyramidal neurons impacted neither the amplitude nor the frequency of mIPSCs [36]. Ryanodine acts as a selective inhibitor of RyRs when administered at micromolar concentrations [18], while at nanomolar concentrations [37], it can trigger RyR opening. In view of the manifold effects on spontaneous release induced by calcium released from RyRs, the presence of specific calcium sensors with various levels of calcium sensitivity on synaptic vesicles has been proposed as a potential explanation [26]. Synchronous release is considered to be mainly triggered by calcium entering through presynaptic VGCCs located at the plasma membrane, as represented by the Ca_v_2.1, 2.2, 2.3 channels [38]. The mechanisms underlying synchronous release entail fast kinetics of calcium-channel activation as well as high local calcium concentrations (up to 100 µM) and close proximity between the calcium source and the calcium sensor [39]. The distance between calcium sensor and calcium source has been estimated to be in the range of 100 nm [26], decreasing up to 10–20 nm in GABAergic synapses [40]. All together, these properties allow the synaptic delay between presynaptic calcium entry and the presynaptic fusion of vesicles containing neurotransmitters within the presynaptic membrane to be reduced. Therefore, the generation of postsynaptic currents that are highly synchronized and characterized by a synaptic delay shorter than 1 ms [40] is guaranteed. RyRs modulate synchronous release and there is growing evidence to indicate that they are distributed at both pre- and postsynaptic sites and that they are involved in the generation of different forms of synaptic plasticity [33,41,42,43,44]. In this regard, a noticeable variation in effects has been observed, with these differences depending on the experimental model and cell type. Though RyRs inhibition significantly affects short-term synaptic plasticity in CA3 pyramidal neurons [33], no effects have been observed in stellate cells [45]. On the other hand, single or sparse APs do not trigger calcium release through RyR activation and, with some exceptions [43], do not significantly contribute to the average amplitude of post synaptic currents [33,45,46]. Similar conclusions were reached in our laboratory, where we observed that RyRs did not contribute significantly to the generation of spontaneous firing (Figure 1a) and single eEPSCs (evoked Excitatory Post Synaptic Currents) amplitude [7] in primary cultured hippocampal neurons (Figure 1e). A possible explanation for this is that the calcium released from RyRs takes part in the glutamate release when neurons are stimulated at a high frequency. Repetitive high frequency stimulation gives rise to increased amounts of asynchronous release, which typically prevails over synchronous release over time. Asynchronous release, along with synchronous release, shares a common pool of readily released vesicles (RRP_tot_, total Readily Releasable Pool) and is characterized by the presence of calcium sensors with a higher sensitivity to calcium [26]. As a consequence, asynchronous release is not characterized by a strong proximity between calcium source and synaptic vesicles [47], and is triggered by calcium bulk that accumulates at presynaptic sites, especially during prolonged and high-frequency stimulation [48]. Asynchronous release is characterized by a wider range of calcium sources, including RyRs [49]. Both GABAergic [50], and glutamatergic synapses [45,51], are characterized by asynchronous release, whose pivotal role in the control of network excitability [26] and the potentiation of neurotransmitter release [52] has been well-documented. Neuronal networks are characterized by a balance between excitatory and inhibitory synapses, where inhibitory synapses represent 80% of the total synaptic conductance [53,54]. The majority of inhibitory synapses belong to cholecystokinin (CCK) and parvalbumin (PV) positive interneurons, with the former mainly characterized by a higher degree of asynchronous release than the latter [55], and by a calcium-dependent mechanism of vesicle release that is mediated by presynaptic N-type (Ca_v_2.2) calcium channels in CCK neurons and P/Q (Ca_v_2.1) channels in PV neurons. The involvement of calcium released from RyRs in the modulation of the asynchronous release of GABAergic synapses has, so far, only been suggested [27]. In this regard, experiments performed on cerebellar basket cells [43] only revealed that RyR inhibition decreases the synchronous release of GABA, while their involvement in the modulation of asynchronous release is still to be elucidated. Regarding glutamatergic synapses, the precise role that RyRs play in asynchronous release has also been speculated [56], but not fully explained. It has been suggested that the asynchronous (or delayed) release of glutamate is not affected by calcium released from RyRs, at least in a context of presynaptic stimuli characterized by low frequency and short duration [45].

## 3. Involvement of RyRs in the Control of Neuronal Excitability

RyRs are involved in the control of the firing patterns [57] and frequency inhibition [58] of thalamocortical neurons as they potentiate the amplitude of the afterhyperpolarization current (I_AHP_) by means of the Ca^2+^-dependent activation of potassium channels (IK_Ca_), which can be measured at the end of the action potential (AP) waveform. However, other evidence suggests that RyRs are functionally coupled to other potassium channels, such as K_v_2.1 [59]. IK_Ca_ can be functionally coupled to VGCCs and contribute to shaping the AP and firing patterns. In this regard, small conductance (SK) potassium channels display a high sensitivity to calcium and do not need to be in close proximity to channels, unlike big conductance (BK) potassium channels [60]. This fascinating feature quite closely resembles the different sensitivities to calcium ions shown by synaptic vesicles and once again paves the way for understating calcium’s manifold mechanisms of action in governing neuronal function. I_AHP_ are temporally distinguished in fast (fAHP), medium (mAHP), and slow (sAHP) AHP, while fAHP is attributed to the opening of BK channels, intermediate and small conductance potassium channels (IK, SK) are responsible for the activation of sAHP and mAHP, respectively [12,61,62]. The calcium source required for IK_Ca_ activation can include RyRs. It has been shown that the calcium released from RyR_3_ in hippocampal CA1 pyramidal neurons is responsible for IK activation and governs firing activity [63]. On the other hand, it has been suggested that BK channels are not activated by the calcium released from RyRs in primary cultured hippocampal neurons, and that they do not contribute to modulating the spontaneous firing of the hippocampal network [7] (Figure 1b), although this functional coupling has been observed and assessed in other neurons [64,65]. Finally, there is no proof, at least under physiological circumstances, that RyRs and mAHP have any functional relationship [62].

## 4. Altered Function of RyR during AD

Moving beyond physiological conditions, increased RyR expression and function have been observed during AD onset, and a relationship between RyR dysfunction and calcium dyshomeostasis has been proposed [66]. Currently available information suggests that the expression of the RyR_2_ and RyR_3_ isoforms is increased in hippocampal neurons from APPPS1 mice [67], while the expression of all three RyRs isoforms is increased in a human neuroblastoma cell line that overexpresses Swedish-mutation-harboring APP [68]. By contrast, only the expression of RyR_3_ is increased in the cortical neurons of TgCRND8 mice [69]. Finally, a functional alteration of RyR_2_ isoforms has been reported as occurring in the brains of AD patients and in two AD murine models. This alteration leads to a leaky outflow of Ca^2+^ from the ER and increased [Ca^2+^]_i_, which is responsible for Ca^2+^ dyshomeostasis and synaptic dysfunction [70]. In spite of these findings, a precise description of how AD affects RyR function is still lacking, probably because of the multiple causes that contribute to generating the disease and the use of different experimental models to study the onset of this pathology. In light of this, we draw the conclusion that, even though the available data indicate a potentiation of the mechanism by which Ca^2+^ is released from RyRs, the presence of a RyRs isoform that is more crucial to initiating this effect is still unknown.

### 4.1. Consequences of RyR Dysfunction on Synaptic Activity during AD

Alzheimer’s disease has been associated with synaptic disruption [71,72], leading to network hyperactivity [73], followed by decreased neuronal activity [74,75]. Nevertheless, the precise changes induced by AD on synaptic properties remain unclear. The conflicting observations depend on several factors, including models, neuronal type, and also the contrasting effects induced by the amyloid beta peptides administered at different concentrations and at different stages of aggregation. Recent data indicate that the glutamate release probability and amplitude of eEPSCs are reduced in 6/7-month old APP/PS1 mice, without changes in the size of the readily releasable pool (RRP) [76]. Similar results have been obtained by treating hippocampal neurons with 400 nM of Abeta42 oligomers [76]. Accordingly, our recent observations suggest that Abeta42 oligomers administered at concentrations ranging between 200 nM and 1 µM decreased spontaneous network firing via the inhibition of glutamatergic synapses. Therefore, it has been suggested that, when glutamatergic synapses are stimulated at high frequencies (above 50 Hz), Abeta42 oligomers increase the amount of calcium released from RyRs. This increased release is, in turn, responsible for the activation of presynaptic BK channels [7], causing presynaptic membrane hyperpolarization, reducing the quantity of calcium entering through presynaptic VGCCs and leads to synaptic and firing inhibition that can be partly restored by dantrolene and paxilline, which are RyR- and BK-channel inhibitors, respectively (Figure 1c,d). The impact of Abeta42 oligomers on glutamatergic synapses has recently been detailed [3,77], revealing that, although the above mechanism is observed at high neuronal firing frequencies, when considering single eEPSCs that are generated via activation of the NMDA receptor, Abeta42 increases their amplitude by increasing the glutamate release probability, the number of release sites, and the size of the readily releasable pool of vesicles available for synchronous release (RRP_syn_) (Figure 2).

Similar results have been observed in a mouse model [78] that is characterized by neuronal hyperexcitability and dependent on an aberrant amount of calcium being released from presynaptic RyRs, which in turn, increases the release probability of glutamatergic synapses. The fact that Abeta42 induces this range of diverse effects may depend not only on the degree of synaptic stimulation in terms of frequencies but also on the progression of the pathology. Therefore, it has been suggested that stimulating effects characterize the early stages of AD, when Abeta amyloid accumulation is lower, while inhibitory effects are more frequently observed during the late stages of the disease [79]. Furthermore, we have recently shown that Abeta42 accumulation at excitatory synapses targets the postsynaptic membrane and reduces the number of NMDA receptors [3] (Figure 2). NMDARs are involved in long-term synaptic plasticity and are responsible for the activation of the CICR mechanism following RyR activation [3]. Their decrease following Abeta42 accumulation may contribute to the impairment of long-term synaptic plasticity and related memory formation during AD onset [80]. The effect of AD onset on GABAergic synapses has also been studied, and an increased number of GABAergic projections towards the CA1 hippocampal region [81] has been observed in APPPS1 mice, together with a potentiation of GABAergic synaptic activity observed in transgenic mice overexpressing hAPP carrying the Swedish and Indiana FAD mutations [82,83]. These results contradict those of other studies, in which AD has been shown to have an inhibitory impact on GABAergic synapses observed in double transgenic APP23xPS45 mice that overexpress both the amyloid precursor protein (APPS_we_) and mutant presenilin 1 [84]. We have recently focused on the effects of Abeta42 oligomers on GABAergic synapses, and observed an increase in eIPSC (evoked Inhibitory Post Synaptic Currents) amplitude together with an increased number of release sites and RRP_syn_ size (dependent on a higher amount of calcium release from RyRs [85]), although the probability of release did not change (Figure 3). These results suggest that the Abeta42 mechanisms of action may be different depending on the synapses involved. We also focused on the effects of Abeta42 on the asynchronous release of GABA and observed that Abeta42 had an inhibitory effect, contradicting the observations made when synchronous release was measured [85]. This result explains our results, which indicated that RRP_tot_ was of comparable size in the controls and the presence of Abeta42 [85] (Figure 3). Therefore, we concluded that the overactivation of RyRs is an example of a recurring feature, although the effects of AD and, in particular, of Abeta42 on synaptic function are multifaced and depend on the type of synapse and stimulation mode (high vs. low frequency stimulation, long vs. short term stimulation).

### 4.2. Consequences of RyR Dysfunction on Neuron Excitability during AD

General hyperexcitability is often observed during AD, although many exceptions have been described [7,62,86]. Early studies performed in 5xFAD mice focused on I_AHP_ [61], and observed that the medium (mAHP) and slow (sAHP) components were reduced and responsible for increased intrinsic excitability. This latter is known to be modulated by calcium released from RyRs [63], but the mechanism behind the recruitment of IK_Ca_ during AD, along with the rise in intrinsic hyperexcitability, is still unclear, and a reduction in firing activity seems to be a more logical consequence of the increased calcium released from RyRs. Another hypothesis suggests that the amount of A-type K^+^ current is decreased in 5xFAD mice [87], and following the administration of Abeta42 oligomers [88], that hyperexcitability is consequently observed. In this regard, it has been suggested that A-type K^+^-channel expression depends on the K_v_4.2 subunit, which in turn, is found to be inhibited by the calcium released from RyRs [87], explaining why an increase in the amount of calcium released from RyRs, caused by the reduced expression of A-type K^+^ channels, is responsible for neuronal hyperexcitability.

### 4.3. Proposed RyR-Inhibition-Based Therapies for AD Treatment

In AD mouse models, it has recently been shown that the accumulation of Abeta oligomers prevents the proper interaction between RyRs and the intracellular stabilizer proteins calstabins, preventing the complete closure of RyRs, which assume a leaky conformational state and generate continuous slow release and accumulation of calcium in the cytoplasm [70]. It has been suggested that this process is associated with the PKA-dependent phosphorylation of RyRs, a pathway that is over activated by amyloid beta peptides via the activation of beta2 adrenoreceptors [89]. Two different isoforms of calstabins are known, and while calstabin1 is preferentially associated with RyR_1_ and RyR_3_, calstabin2 has a higher affinity for RyR_2_ [90]. In our laboratory, we have recently shown that RyR inhibition, via the administration of dantrolene, counteracts the increases in [Ca^2+^]_i_ that are induced by Abeta42 and improves the excitable and synaptic properties of hippocampal neurons [3,7] (Figure 1c and Figure 2). Dantrolene is actually the only RyRs inhibitor (RyRI) used for the clinical treatment of malignant hyperthermia [91], but its use for AD treatment is not conceivable due to problems related to stability, selectivity, and permeability through the blood–brain barrier (BBB) [92]. New classes of RyRIs are being studied, and among those of particular interest, we can find the class of RyCal compounds (RyCals) [70,85,89], which can promote the binding between calstabins and RyRs. The only commercially available RyCal molecule is S107, but data on its effect on neuronal function are scarce. In this context, we have recently shown that S107 counteracts the effects induced by Abeta42 by reducing eIPSC amplitude to values comparable to those observed under control conditions (Figure 3). Moreover, the unbalance between synchronous and asynchronous release is also recovered by s107 administration [85] (Figure 3). Interesting results have been obtained through the administration of carvedilol [87,93]. This compound is a racemic mixture of s and r carvedilol, and while the S-enantiomer is an adrenoreceptor inhibitor, the R-enantiomer inhibits RyRs activity without any affinity for beta adrenergic receptors [93]. In view of the cardiovascular effects caused by the inhibition of adrenergic receptors, there has been a preference for using R-carvedilol in the treatment of impairments induced by AD onset. Interestingly, a significant rescue of neuronal function following R-carvedilol administration has been observed in 5xFAD mice, with this effect not being detectable in the presence of the racemic mixture [87]. On the other hand, promising results have also been achieved through the inhibition of beta_2_ adrenoreceptors [89], indicating that perhaps the racemic mixture of carvedilol should be used. In conclusion, there is no clear evidence as to whether the racemic mixture or single enantiomer of carvedilol are preferable for the targeting of RyRs and related synaptic dysfunction. Moreover, the question of whether the inhibition of beta2 adrenoreceptors may represent a promising strategy for AD treatment is still yet to be satisfactorily answered.

## 5. Conclusions

Studies performed in recent years indicate that RyRs may play a role in the onset of AD and other neurodegenerative diseases [94]. The development of pharmacological therapy is far from being established because different RyR isoforms, of which the exact function is partly unknown due to a lack of selective RyRs-isoform inhibitors, are present in the brain. Additionally, it is necessary to consider their principal roles in other organs, including the heart and arteries. While there is considerable evidence to suggest that the inhibition of RyRs might protect neuronal function during AD, much remains to be done, especially in the understanding of the mechanisms induced by RyR dysfunctions that lead to alterations in synaptic properties and the intrinsic excitability of central neurons during AD onset.

## Figures and Tables

**Figure 1 cells-12-01236-f001:**
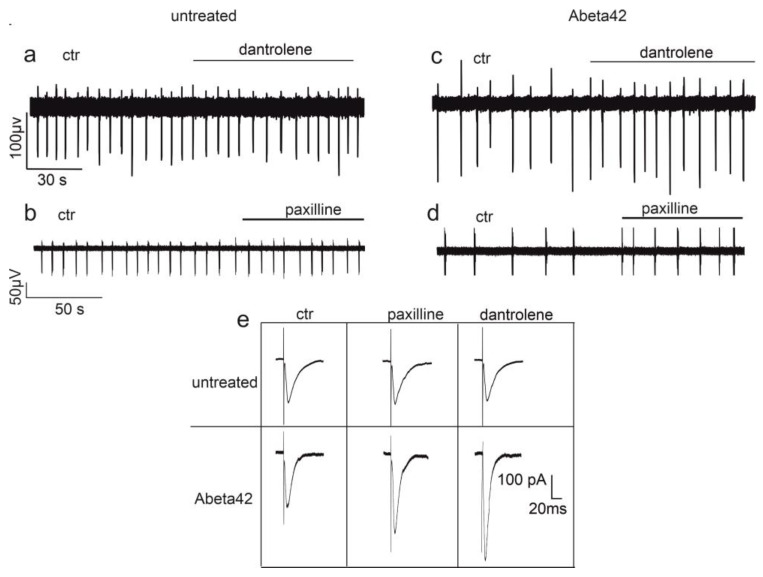
Abeta42 inhibits hippocampal network excitability by targeting RyRs. (**a**)Spontaneous electrical activity of primary cultured hippocampal network, as recorded by MEA (one representative trace), showing that, in the absence of Abeta42 (ctr), RyR inhibition through dantrolene does not affect the spontaneous frequency of extracellularly recorded action potentials. (**b**) Similar results have been obtained by blocking BK (big conductance potassium) channels using paxilline, suggesting that, under control conditions, neither the RyR nor BK channels are involved in the modulation of hippocampal network excitability. (**c**,**d**) When neurons are incubated with Abeta42 oligomers, it is observed that decreased firing activity is partly rescued by dantrolene or paxilline administration. (**e**) eEPSC amplitude is increased following paxilline or dantrolene administration in hippocampal neurons previously treated with Abeta42, while no effects were observed in the absence of Abeta42 (untreated neurons) adapted from [7].

**Figure 2 cells-12-01236-f002:**
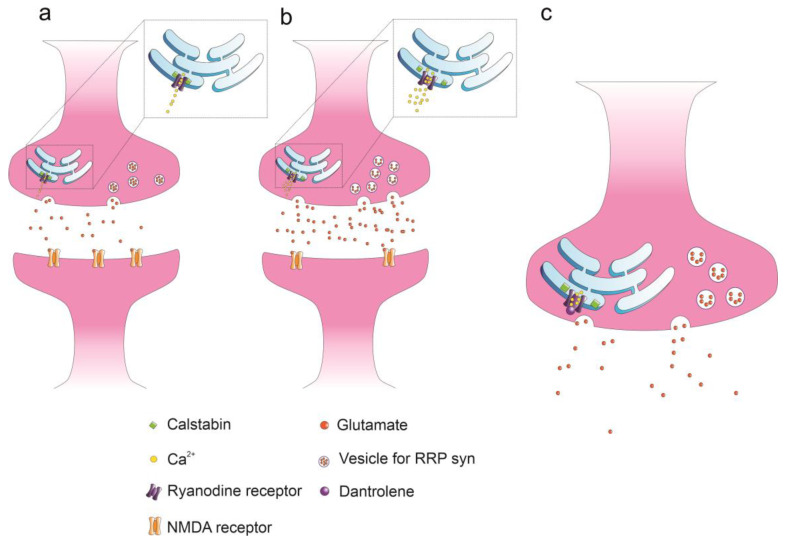
Glutamatergic eEPSCs that are dependent on the activation of the NMDA receptor are potentiated by Abeta42. (**a**,**b**) Abeta42 increases the amount of calcium released from RyRs by potentiating the release probability associated with glutamate release, increasing the size of the readily releasable pool associated with synchronous release (RRP_syn_), and the number of release sites, but postsynaptically decreases the number of synaptic NMDA receptors. (**c**) Dantrolene, an allosteric inhibitor of RyRs, restores the synaptic properties observed under control conditions (**a**).

**Figure 3 cells-12-01236-f003:**
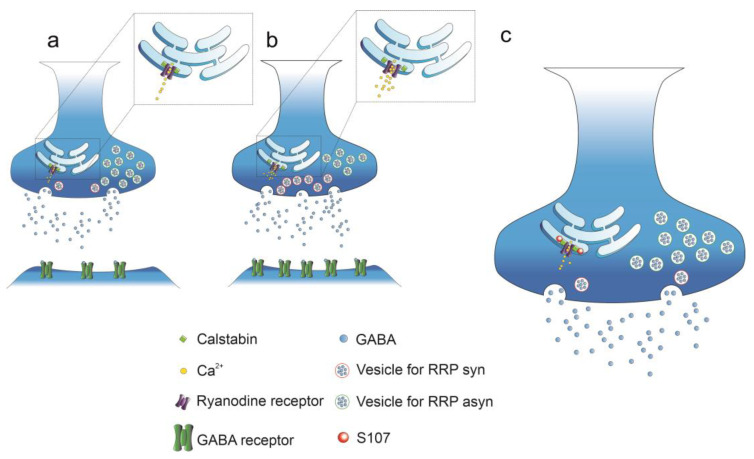
GABAergic eIPSCs are potentiated by Abeta42. (**a**,**b**) Abeta42 increases the amount of calcium released from RyRs and the number of release sites. The total readily releasable pool (RRP_tot_) size associated with synchronous and asynchronous release (RRP_syn_ and RRP_asyn_) and the release probability associated with GABA release are not affected by Abeta42, while increased RRP_syn_ size is observed together with decreased RRP_asyn_ size. At the postsynaptic site, Abeta42 increases the number of GABA_A_ receptors. (**c**) The administration of the RyRs-calstabin interaction stabilizer S107 restores the synaptic properties observed under control conditions (**a**).

## Data Availability

No new data were created or analyzed in this study. Data sharing is not applicable to this article.

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
