# Peer review of "The Role of Ryanodine Receptors in Regulating Neuronal Activity and Its Connection to the Development of Alzheimer’s Disease"

_cells, 2023, doi:10.3390/cells12091236_

Round 1

Reviewer 1 Report

The manuscript is well-organized and reveals interesting novel approaches to cerebral RyRs in physiology and AD pathology.

Although the quality of the paper deserves publication here are some minor comments to assess:

1. concepts such as mEPSCs, mIPSCs, manyfold effects, AP, eIPSCs, A-type K+ should be at least written in extenso the first time or briefly described for the general audience

2. BK abbreviation in figure 1 caption should be explained, as has been done in the main text.

3. Carefully check the manuscript for typos

Author Response

The manuscript is well-organized and reveals interesting novel approaches to cerebral RyRs in physiology and AD pathology.

Although the quality of the paper deserves publication here are some minor comments to assess:

  • concepts such as mEPSCs, mIPSCs, manyfold effects, AP, eIPSCs, A-type K+ should be at least written in extenso the first time or briefly described for the general audience

We thank the reviewer for this suggestion. We added the necessary explanations for the abbreviations

  • BK abbreviation in figure 1 caption should be explained, as has been done in the main text.

We explained the abbreviation as requested by the referee

  • Carefully check the manuscript for typos/ Countless grammar mistakes that make it difficult to understand

The paper has been edited by a native English-speaker. We think it is more satisfactory than the last version

Reviewer 2 Report

  • Countless grammar mistakes that make it difficult to understand.

  • Figure 1: There is no label “a)” in the description to describe its corresponding graph

  • Figure 3 a,b: The description and illustration do not match. The description states that Abeta 42 increases the size of RRPsyn and decreases the size of RRPasyn. In the illustration, the size of RRPsyn decreases and the size of RRPasyn increases.
  • This is not an appropriate review, lot of mistakes and plagiarism

Author Response

  • Figure 1: There is no label “a)” in the description to describe its corresponding graph

We carefully verified the text and found through the text the description of Fig.1a

  • Figure 3 a,b: The description and illustration do not match. The description states that Abeta 42 increases the size of RRPsyn and decreases the size of RRPasyn. In the illustration, the size of RRPsyn decreases and the size of RRPasyn

We are grateful to the reviewer for this comment. The mistake has been corrected

  • This is not an appropriate review, lot of mistakes and plagiarism

We have deemed it important to demonstrate the seriousness of our working group and to underline the commitment with which the manuscript was written and we checked the plagiarism by using iThenticate, a plagiarism detection software. The rate of citation from the literature is approximately 11%, and 10% of that refers to our previous studies.

Reviewer 3 Report

The present manuscript reviews the current knowledge on the potential role of RyRs in Alzheimer's disease. The manuscript is well-structured and, although it covers an ill-defined topic, the current evidence already justifies such a review. 

Major:

Figure 2 should also reflect the role of RyRs in Abeta effect at high-frequency stimulations.

Minor issues:

-  there should be a space between values and units

- it would be better to use the symbol beta

- In the figures it should be removed Fig. 1, Fig.2 and Fig.3

 --Specific Comments

1. What is the main question addressed by the research?
-Review the role of RYRs in Alzheimer's disease. Potential value of drug-targeting RYRs.

2. Do you consider the topic original or relevant in the field? Does it
address a specific gap in the field?
-I don´t think it is a very exciting topic but the current knowledge justifies a review. I am not sure if it justifies being published in Cells, but that's your editorial decision.

3. What does it add to the subject area compared with other published
material?
-There are two recent reviews in. J Clin Inv. and in a Frontiers journal reviewing RYRs in different disorders including Alzheimer's disease. But just focused on Alzheimer's disease, the last review was published in 2020. There is no great novelty. Just an update.

4. What specific improvements should the authors consider regarding the
methodology? What further controls should be considered?
-It does not apply, since it is a review. But I find it well-structured.

5. Are the conclusions consistent with the evidence and arguments presented and do they address the main question posed?
-There is lots of speculation, but it is based on evidence. I think it is acceptable.

6. Are the references appropriate?
-It seems so.

7. Please include any additional comments on the tables and figures.
-I already provided suggestions on the figures.

Author Response

The present manuscript reviews the current knowledge on the potential role of RyRs in Alzheimer's disease. The manuscript is well-structured and, although it covers an ill-defined topic, the current evidence already justifies such a review.

Major:

  • Figure 2 should also reflect the role of RyRs in Abeta effect at high-frequency stimulations.

We appreciate the reviewer's input, but more research is needed to fully understand the precise function of RyRs during high-frequency stimulation. For this reason we did not mention it at Fig.2.

Minor issues:

  • there should be a space between values and units

We thank the reviewer for this comment. We added a space between values and units.

  • it would be better to use the symbol beta

We thank the reviewer's feedback. In this regard, we chose to refrain from using symbols in keeping with our most recent publications from the previous years.

  • In the figures it should be removed Fig. 1, Fig.2 and Fig.3

We followed the reviewer’s suggestion

Reviewer 4 Report

The authors have attempted to review literature supporting the role of ryanodine receptors (RyRs) in the progression of neurodegenerative states. While many of the points are present in the manuscript - there is a serious lack of flow in this version of the manuscript that prevents appreciation of the subject matter. The sentences have been placed before and then explained later, which does not work very well. It is very disheartening that the first sentence of introduction does not make any sense. These are serious issues that need to be addressed.

Author Response

The authors have attempted to review literature supporting the role of ryanodine receptors (RyRs) in the progression of neurodegenerative states. While many of the points are present in the manuscript - there is a serious lack of flow in this version of the manuscript that prevents appreciation of the subject matter. The sentences have been placed before and then explained later, which does not work very well. It is very disheartening that the first sentence of introduction does not make any sense. These are serious issues that need to be addressed.

We carefully verified the text accordingly to what the reviewer attempted to suggest. The manuscript has been revised by an English native speaker editor. We now are confident that the quality of the paper has been significantly improved.

Round 2

Reviewer 4 Report

Some issues need attention:

1) Line 85 - IP3Rs is not introduced before or elaborated later.

2) Line 205 onwards - Beautiful explanation on how AD and AD-like mouse models show similarities in the dysregulation of RyRs, but differences in isoforms and this is an important avenue that should be emphasized again in the conclusion and as a pictorial representation - this is a unique compendium of the finding that the authors deserve credit for.

3) Line 267 - please state the name of the mouse model for the benefit of the readers.
